# Level-Set-Based Kidney Segmentation from DCE-MRI Using Fuzzy Clustering with Population-Based and Subject-Specific Shape Statistics

**DOI:** 10.3390/bioengineering9110654

**Published:** 2022-11-05

**Authors:** Moumen El-Melegy, Rasha Kamel, Mohamed Abou El-Ghar, Norah S. Alghamdi, Ayman El-Baz

**Affiliations:** 1Electrical Engineering Department, Assiut University, Assiut 71515, Egypt; 2Computer Science Department, Assiut University, Assiut 71515, Egypt; 3Radiology Department, Urology and Nephrology Center, Mansoura University, Mansoura 35516, Egypt; 4Department of Computer Sciences, College of Computer and Information Sciences, Princess Nourah bint Abdulrahman University, P.O. Box 84428, Riyadh 11671, Saudi Arabia; 5Bioengineering Department, University of Louisville, Louisville, KY 40292, USA

**Keywords:** DCE-MRI, fuzzy c-means, kidney segmentation, level set, statistical shape models, U-Net

## Abstract

The segmentation of dynamic contrast-enhanced magnetic resonance images (DCE-MRI) of the kidney is a fundamental step in the early and noninvasive detection of acute renal allograft rejection. In this paper, a new and accurate DCE-MRI kidney segmentation method is proposed. In this method, fuzzy c-means (FCM) clustering is embedded into a level set method, with the fuzzy memberships being iteratively updated during the level set contour evolution. Moreover, population-based shape (PB-shape) and subject-specific shape (SS-shape) statistics are both exploited. The PB-shape model is trained offline from ground-truth kidney segmentations of various subjects, whereas the SS-shape model is trained on the fly using the segmentation results that are obtained for a specific subject. The proposed method was evaluated on the real medical datasets of 45 subjects and reports a Dice similarity coefficient (DSC) of 0.953 ± 0.018, an intersection-over-union (IoU) of 0.91 ± 0.033, and 1.10 ± 1.4 in the 95-percentile of Hausdorff distance (HD95). Extensive experiments confirm the superiority of the proposed method over several state-of-the-art level set methods, with an average improvement of 0.7 in terms of HD95. It also offers an HD95 improvement of 9.5 and 3.8 over two deep neural networks based on the U-Net architecture. The accuracy improvements have been experimentally found to be more prominent on low-contrast and noisy images.

## 1. Introduction

Acute rejection is the most frequent cause of graft failure after kidney transplantation [1]. However, acute renal rejection is treatable, and early detection is critical in order to ensure graft survival. The diagnosis of renal transplant dysfunction using traditional blood and urine tests is inaccurate because the failure can be detected after losing 60% of the kidney function [1]. In this respect, the DCE-MRI technique has achieved an increasingly important role in measuring the physiological parameters of the kidney and follow-up patients. DCE-MRI data acquisition is carried out through injecting the patient with a contrast agent and, during the perfusion, the kidney images are captured quickly and repeatedly at three second intervals. The contrast agent perfusion leads to contrast variation in the acquired images. Consequently, the intensity of the images at the beginning of the sequence is low (pre-contrast interval), gradually increases until reaching its maximum (post-contrast interval), and then decreases slowly (late-contrast interval). Figure 1 shows a time sequence of DCE-MRI kidney images of one of the patients that was taken during the contrast agent perfusion. Accurate kidney segmentation from these images is an important first step for a complete noninvasive characterization of the renal status. However, segmenting the kidneys is challenging due to the motion that is made by the patient’s breathing, the contrast variation, and the low spatial resolution of DCE-MRI acquisitions [1,2].

In order to overcome these problems, several researchers have proposed multiple techniques to segment the kidney from DCE-MRI images. A careful examination of the related literature reveals that level-set-based segmentation methods [3,4,5,6,7,8] have been the most popular for this purpose. In these methods, a deformable model adapts to the shape of the kidney, its evolution being constrained by the image properties and prior knowledge of the expected kidney shape. In [3], the authors developed a DCE-MRI kidney segmentation method employing prior kidney shape and gray-level distribution density in the level set speed function in order to constrain the evolution of the level set contour. However, their method had large segmentation errors on noisy and low-contrast images. Thus, in [4,5], Khalifa et al. proposed a speed function combining the intensity information, the shape prior information, and the spatial information modeled by 2nd- and 4th-order Markov Gibbs random field (MGRF) models, respectively. In order to circumvent the issue of the rather similar appearance between the kidney and the background tissues, Liu et al. [6] proposed to remove the intensity information from the speed function in [5] and to use a 5th-order MGRF to model the spatial information.

Incorporating the shape information into the level set method typically requires a separate registration step [3,4,5,6] to align an input DCE-MRI image to the shape prior model in order to compensate for the motion that is caused by the patient’s breathing and movement during data acquisition. In a different manner, Hodneland et al. [7] proposed a new model that jointly combines the segmentation and registration into the level set’s energy function and applied it to segment kidneys from 4D DCE-MRI images.

From another perspective, the level set contour evolution is guided by deriving a partial differential equation in the direction that minimizes a predefined cost functional containing several weighting parameters that need manual tuning [3,4,5,6,7]. In contrast, Eltanboly et al. [8] proposed a level set segmentation method employing the gray-level intensity and shape information without using weighting parameters. Some work [9,10] has also been carried out in addressing the intensity inhomogeneity and the low contrast problems of DCE-MRI images that are caused during the acquisition process. Based on fractional calculus, Al-Shamasneh et al. [9] proposed a local fractional entropy model to enhance the contrast of DCE-MRI images. Later, in [10], they presented a fractional Mittag-Leffler energy function based on the Chan-Vese algorithm for segmenting the kidneys from low-contrast and degraded MR images.

More recently, convolutional neural networks (CNNs) have been successfully used for several image segmentation tasks, including kidney segmentation. For example, Lundervold et al. [11] developed a CNN-based approach for segmenting kidneys from 3D DCE-MRI data using a transfer learning technique from a network that was trained for brain hippocampus segmentation. Haghighi et al. [12] employed two cascaded U-Net models [13] to segment kidneys from 4D DCE-MRI data. Later on, Milecki et al. [14] developed a 3D unsupervised CNN-based approach for the same reason. Bevilacqua et al. [15] presented two different CNN-based approaches for accurate kidney segmentation from MRI data. On the other hand, the authors in [16] integrated a mono-objective genetic algorithm and deep learning for an MRI kidney segmentation task. Isensee et al. [17] presented the top scoring model in the CHAOS challenge [18] for an abdominal organs segmentation task, in which they used an nnU-Net model to segment the left and right kidneys from MRI data. The CHAOS challenge dataset includes the data of 80 different subjects, including 40 CTs and 40 MRIs. Each sequence contains an average of 90 scans in CT and 36 in MRI in the DICOM format.

The research gap is as follows: The common stumbling block facing CNN methods is that they typically require annotated data of a large size in order to train the network, which is often difficult to obtain in the medical field. Thus, the aforementioned deep learning methods struggle to achieve high segmentation accuracy. On the other hand, the level-set-based kidney segmentation methods [3,4,5,6,7,8] have proved their effectiveness in achieving a superior performance with more accurate segmentation. However, unfortunately, almost all of them need accurate level set contour initialization to be performed manually by the user. Inaccurate initialization may cause a drop in the segmentation accuracy or even cause the method to fail. In order to overcome this problem, in [19] we have presented an automated DCE-MRI kidney segmentation, called FCMLS, based on FCM clustering [20] and level sets [21]. In our FCMLS method, we constrain the contour evolution by the shape prior information and the intensity information that are represented in the fuzzy memberships. In addition, in order to ensure the robustness of the FCMLS method against contour initialization, we employ smeared-out Heaviside and Dirac delta functions in the level set method. The FCMLS method has indeed demonstrated its efficiency in segmenting the kidneys from DCE-MRI images. However, it still has some limitations. First, its performance drops on low-contrast images, such as those in the pre- and late-contrast parts of the time sequence in Figure 1. Second, the FCM algorithm is used for computing the fuzzy memberships of the image pixels before the level set evolution begins. Once the level set starts evolving, the obtained memberships are not changed, and this might be not accurate enough in some cases.

In order to enhance the segmentation accuracy of FCMLS, and to improve its robustness on low-contrast images, we have developed a new kidney segmentation method, named the FML method, in [22]. In this method, we model the correlation between neighboring pixels into the level set’s objective functional by a Markov random field energy term. We also embed the FCM algorithm into the level set method and iteratively update the fuzzy memberships of the image pixels during contour evolution. The experimental results have confirmed the improved accuracy and robustness of this method. However, the integration of the Markov random field model within the level set formulation has increased the computational complexity of the FML method significantly.

In this paper, we follow a different strategy in order to improve the segmentation performance of our previous method without sacrificing the computational complexity. The shape information plays a key role in kidney segmentation since human kidneys tend to have a common shape, with between-subject variations. Thus, we seek to take full advantage of this in our new level set formulation by exploiting the level set method’s flexibility to accommodate the shape information about the target object that is to be segmented [23]. Inspired by [24], we employ PB-shape and SS-shape models for kidney segmentation. The PB-shape model is built offline from a range of kidney images from various subjects that are manually segmented by human experts, whereas the SS-shape model is constructed on the fly from the segmented kidneys of a specific patient.

This new methodology is able to generate high segmentation accuracy because the PB-shape model is used on images with high contrast in the post-contrast interval of the image sequence. Moreover, the SS-shape model that is generated from those accurate segmentations is employed on the more challenging, lower contrast images from pre- and late-contrast intervals of the sequence, as it more accurately reflects the kidney’s shape from the same patient. Our early work on this new methodology has been drafted in [25], on which we build and develop several novel contributions in the present paper. First, we embed FCM clustering into the level set evolution. Thus, the kidney/background fuzzy memberships are computed and updated every time the level set contour evolves. Second, the representation of the shape information in [25] is based on a 1st-order shape method, which might be inaccurate when some kidney pixels are not observed at all in the images that are used to construct the shape model. In this paper, we adopt an efficient Bayesian parameter estimation method [26] in computing the PB-shape and SS-shape models, which more accurately accounts for the kidney pixels that are possibly not observed during the model building. Third, we propose an automated and time-efficient, yet effective, strategy to determine the images from the patient’s sequence, to which the PB-shape model, the SS-shape model, or both of the models blended together are applied.

The proposed method is used to segment the kidneys of 45 subjects from DCE-MRI sequences, and the segmentation accuracy is assessed using the Dice similarity coefficient (DSC), the intersection-over-union (IoU), and the 95-percentile of Hausdorff distance (HD95) metrics [2,27]. Our experimental results prove that the proposed method can achieve high accuracy, even on noisy and low-contrast images, with no need for tuning the weighting parameters. The experiments also show that the segmentation accuracy is not affected by changing the position of the initial level set contour, which demonstrates the high consistency of the proposed method. We compare our method’s segmentation accuracy with several state-of-the-art level set methods, as well as our own earlier methods [19,22,25]. Furthermore, we compare its performance against the base U-Net model and one of its modifications named BCDU-Net [28], which is trained for the same kidney segmentation task. The two networks are trained from scratch on our DCE-MRI data, which are augmented with the KiTS19 challenge dataset [29]. This dataset contains 300 subjects’ data, where 210 out of all of the data are publicly released for training and the remaining 90 subjects are held out for testing. Each subject has a sequence of high quality CT scans, with their ground-truth labels that are manually segmented by medical students. It also includes a chart review that illustrates all of the relevant clinical information about this patient. All of the CT images and segmented annotations are provided in an anonymized NIFTI format. The comparison results confirm that the proposed method outperforms all of the other methods.

The remainder of this paper is structured as follows: Section 2 introduces the mathematical formulation of the proposed kidney segmentation method. Then, Section 3 provides the experimental results and the comparisons. Finally, a discussion and the conclusions are presented in Section 4.

## 2. Materials and Methods

In this section, we present the formulation of the proposed segmentation method in detail.

### 2.1. Materials

DCE-MRI data are collected from 45 subjects who underwent kidney transplantation at Mansoura University Hospital, Egypt. In order to acquire the data, a dose of 0.2 mL/kgBW of Gd-DTPA contrast agent was injected intravenously at a rate of 3–4 mL/s. Meanwhile, the kidney is scanned quickly and repeatedly, at 3 s intervals, using a 1.5T MRI scanner with a phased-array torso surface coil. The transition of the contrast agent results in a variation in the contrast of the images. Therefore, each subject has a dataset of about 80 repeated temporal frames, which are 256 × 256 pixels in size. Each image in the sequence is manually segmented by an expert radiologist at the hospital. A sample sequence of one subject is shown in Figure 1.

### 2.2. Problem Statement and Notations

In DCE-MRI, to evaluate the transplanted kidney function, the kidney needs to be accurately segmented from each image separately. Let It, t=1,…,ℕ, be a time-point image captured at time t from a DCE-MRI sequence of length ℕ. Itx,y is the intensity of a pixel x,y in the image domain Ω. The target is to label each pixel x,y in the image as kidney (K) or background (B).

### 2.3. Level-Set-Based Segmentation Model with Fuzzy Clustering and Shape Statistics

Given a DCE-MRI time-point image $I_{t}\;$, the level set contour ∂Ω partitions the domain Ω of the image into a kidney region $\Omega^{K}$ and background region $\Omega^{B}$. At any time t, ∂Ω corresponds to the level set of a higher-dimensional function $\phi_{t}\left(x,y\right)$, i.e., $\partial \Omega \left(t\right)=\{\left(x,y\right)\;|\;\phi_{t}\left(x,y\right)=0\}$. The function ϕ is defined as the shortest Euclidean distance between every pixel $\left(x,\;y\right)$ in the image and the contour. The distance is positive for the pixels inside of the contour, negative outside, and zero on the contour. The level set contour iteratively evolves in the direction minimizing the following energy function:(1)$$E\left(\phi \left(x,y\right)\right)=\lambda_{1}\;L\left(\phi \left(x,y\right)\right)+\lambda_{2}E_{FCM}\left(\phi \left(x,y\right)\right)$$
where $\lambda_{1}$ and $\lambda_{2}$ are positive normalizing parameters that control the impact of the energy terms. $E_{FCM}\left(\phi \left(x,y\right)\right)$ is an FCM-based energy function computed from the input image $I_{t}$ to attract the contour towards the position of the kidney in the image, which is defined as follows:(2)$$E_{FCM}\left(\phi \right)=\int_{\Omega }^{\;}H\phi_{\varepsilon }\;F_{B}\left(x,y\right)\;dxdy+\int_{\Omega }^{\;}\left[1-H\phi_{\varepsilon }\right]\;F_{K}\left(x,y\right)\;dxdy$$
where $H\phi_{\varepsilon }=H_{\varepsilon }\left(\phi \left(x,y\right)\right)$ is the smeared-out Heaviside function, which is defined as follows:(3)$$H\phi_{\varepsilon }=\left\{\begin{array}{ll} 1 & \phi >\varepsilon \\ \frac{1}{2}+\frac{\phi }{2\varepsilon }+\frac{1}{2\pi }\sin\left(\frac{\pi \phi }{\varepsilon }\right) & -\varepsilon \leq \phi \leq \varepsilon \\ 0 & \phi <-\varepsilon \end{array}\right.$$
where the parameter ε determines the degree of smearing. $L\left(\phi \right)$ in (1) is a length term that is responsible for keeping the level set contour $\phi \left(x,y\right)$ smooth and defined, as follows:(4)$$L\left(\phi \left(x,y\right)\right)=\overset{\;}{\underset{\Omega }{\int }}\delta \phi_{\varepsilon }\;\left|\nabla \phi \left(x,y\right)\right|\;dx\;dy\;\;$$
where $\delta \phi_{\varepsilon }=\delta_{\varepsilon }\left(\phi \left(x,y\right)\right)$ is the Dirac delta function, which is the derivative of $H\phi_{\varepsilon }$, and is given as follows:(5)$$\delta \phi_{\varepsilon }=\left\{\begin{array}{ll} 0 & \left|\phi \right|>\varepsilon \\ \frac{1}{2\varepsilon }+\frac{1}{2\varepsilon }\cos\left(\frac{\pi \phi }{\varepsilon }\right) & \left|\phi \right|\leq \varepsilon \;\;\;\;\; \end{array}\right.$$

$F_{L}\left(x,y\right)$ in (2) represents either kidney (for L=K) or background (for L=B) energy function of the pixel $\left(x,\;y\right)$ in the image and is defined as follows:(6)$$F_{L}\left(x,y\right)=\;\omega_{t}\;\mu_{L}\left(x,y\right)\;P_{L}\left(x,y\right)+\left(1-\omega_{t}\right)\;\mu_{L}\left(x,y\right)\;S_{L}\left(x,y\right)$$
where $\mu_{L}\left(x,y\right)$ is the kidney/background fuzzy membership degrees of the pixel $\left(x,y\right)$.$\;P_{L}\left(x,y\right)$ and $S_{L}\left(x,y\right)$ are prior probabilities of the pixel $\left(x,y\right)$ derived from PB-shape and SS-shape models, respectively. The weight factor $\omega_{t}$ is used to control the contribution of both models in the segmentation operation. Information about how the value of $\omega_{t}$ is computed for each image in the sequence is explained in Section 2.6.

According to the calculus of variations, the minimization of the function in (1), with respect to ϕ, is given as follows:(7)$$\frac{\partial \phi }{\partial t}=\delta \phi_{\varepsilon }\left[\;\lambda_{1}\mathrm{div}\left(\frac{\nabla \phi }{\left|\nabla \phi \right|}\right)+\lambda_{2}\;F_{K}\left(x,y\right)-\lambda_{2}\;F_{B}\left(x,y\right)\;\right]$$

Finally, the level set contour is iteratively evolved to the boundary of the object as follows:(8)$$\phi_{n+1}\left(x,y\right)=\phi_{n}\left(x,y\right)+\tau \frac{\partial \phi_{n}\left(x,y\right)}{\partial t}$$
where integer n is a number of time steps, as follows: t=nτ for τ>0. It is worth noting that using the smeared-out Heaviside and Dirac delta function is important in order to obtain a global minimizer for the function in (1), irrespective of the level set initialization in the image [21].

### 2.4. FCM Membership Function

Given an image $I_{t}$, the FCM clustering algorithm divides the pixels in the image domain Ω into two separate clusters, kidney and background, as shown in Figure 2. According to this algorithm, the optimal centroid values of the clusters and the corresponding membership degrees are obtained by iteratively minimizing an objective function of the following form [30]:(9)$$J=\overset{\;}{\underset{\left(x,y\right)\;\in \;\Omega \;}{\sum }}\overset{\;}{\underset{L}{\sum }}\;{\mu_{L}}^{2}\left(x,y\right)\;||I_{t}\left(x,y\right)-C_{L}||{}^{2}$$
where $C_{L}$ is the centroid value of kidney (L=K) or background (L=B) clusters, and $\mu_{L}\left(x,y\right)\in \left[0,1\right]$ is the fuzzy membership degree of the pixel $\left(x,y\right)$ in the cluster L and satisfies the condition of $\mu_{K}\left(x,y\right)+\mu_{B}\left(x,y\right)=1$. || .|| represents the Euclidean distance between the pixel’s intensity and cluster’s centroid.

In our earlier method [25], the FCM algorithm is used to compute the fuzzy memberships of the image pixels before the level set evolution begins. Once the level set starts evolving, the obtained memberships are not changed, and this might be not accurate enough in some cases. We improve this approach in the present paper. First, the kidney and background centroid values are initially defined as the mean of the pixel intensities inside and outside of the initial level set contour, respectively. Then, the kidney/background fuzzy membership degrees of each pixel $\left(x,y\right)$ are iteratively updated during the level set evolution as follows:(10)$$\mu_{L}\left(x,y\right)=\frac{||I_{t}\left(x,y\right)-C_{L}||{}^{-2}}{||I_{t}\left(x,y\right)-C_{K}||{}^{-2}+||I_{t}\left(x,y\right)-C_{B}||{}^{-2}}$$

Similarly, the centroid values of the kidney/background clusters are computed as follows:(11)$$C_{L}=\frac{\;\sum_{\left(x,y\right)\in \Omega }\;\;R_{L}\left(\phi \left(x,y\right)\right)\;\;I_{t}\left(x,y\right)\;\;\;{\mu_{L}}^{2}\left(x,y\right)\;\;}{\sum_{\left(x,y\right)\in \Omega }\;R_{L}\left(\phi \left(x,y\right)\right)\;\;{\mu_{L}}^{2}\left(x,y\right)\;\;}$$
where $R_{L}\left(\phi \left(x,y\right)\right)=R_{K}\left(\phi \left(x,y\right)\right)=H\phi_{\varepsilon }$ for (L=K), and $R_{L}\left(\phi \left(x,y\right)\right)=R_{B}\left(\phi \left(x,y\right)\right)=\left(1-H\phi_{\varepsilon }\right)$ for (L=B). As such, the per-pixel fuzzy memberships and kidney/background centroids are coupled with the level set function via (10) and (11) and are updated in each evolution step.

Overall, the membership values of the pixels to a specific cluster depend on the distances between the intensity of the pixels and the cluster centroid. This means that the pixels are assigned high membership values (close to 1) to a certain cluster when their intensities are close to the centroid value and low membership values (close to 0) when they are far from the centroid. As illustrated in Figure 2, the higher the brightness of a pixel is in the kidney/background cluster, the higher its probability to belonging to this cluster. As shown in Figure 2, relying only on fuzzy membership is often not enough to obtain accurate kidney segmentation, especially on low-contrast images. Thus, we incorporate the shape prior information with fuzzy memberships to control the level set evolution.

### 2.5. Statistical Kidney Shape Model

Some earlier approaches (e.g., [6,25]) employ a 1st-order shape method in the construction of a kidney shape model. The major drawback of this method appears when a pixel is classified as kidney or background in all of the training images. In such cases, the pixel-wise probability of the observed label will be exactly 1, and the unobserved label’s probability will be exactly 0, which is often unreasonable. To circumvent this issue, we adopt the Bayesian parameter estimation method [26] in the construction of the PB-shape and SS-shape models in our work here. For the PB-shape model, a number N of DCE-MRI kidney images are selected from varying subjects, and one among them is considered as a reference image. These images are mutually registered to the selected target image, assuming 2D affine transformation by the maximization of mutual information [31] (Figure 3). Then, the co-aligned images are manually segmented by an expert. Finally, the obtained ground-truth segmentations are used to build the shape model, as follows:

For each pixel $\left(x,y\right)$ in the co-aligned ground-truth images, when kidney and background labels are both observed, the empirical kidney/background probability of this pixel is computed as follows [26,32]:(12)$$P_{L}\left(x,y\right)=\left[\frac{N_{L}\left(x,y\right)+\beta }{\;\;N+\beta \;O\left(x,y\right)\;}\right]\;\left[\frac{N}{\;N+𝓁-O\left(x,y\right)\;}\right]$$
where $O\left(x,y\right)$ denotes the number of observed labels, which in this case equals 2, because both labels are observed. $N_{L}\left(x,y\right)$ indicates how many times the label L is observed. β is a pseudo count added to the count of each observed label, and 𝓁 is the total number of possible region labels (kidney and background). On the other hand, when a kidney or background label is observed in all of the images, $O\left(x,y\right)$ equals 1 and the probability of the observed label is computed from (12), while the probability of the unobserved label is computed as follows:(13)$$P_{L}\left(x,y\right)=\left[\frac{1}{\;\;𝓁-O\left(x,y\right)\;}\right]\;\left[1-\frac{N}{\;N+𝓁-O\left(x,y\right)\;}\right]$$

According to the above steps, an example PB-shape model is shown in Figure 3. The same methodology is also adopted in the construction of the SS-shape model, but from a set of images selected on the fly from the specific patient’s sequence being segmented.

### 2.6. Sequence Partitioning and the Weight Factor

In order to segment the kidney of a specific patient, we partition the patient’s DCE-MRI sequence into three subsets. The already-constructed PB-shape model is employed to segment the kidneys from the images in the first subset $S_{1}$. The obtained kidney segmentations are used to construct the SS-shape model, which is blended with the PB-shape model to segment the kidneys from the images in the second subset $S_{2}$. The images in the third subset $S_{3}$ are segmented using only the SS-shape model.

We propose to employ an automated, fast approach for this sequence partitioning. First, all of the images in a given patient sequence are co-aligned via affine transformations to the reference image used in the PB-shape model construction. Then, for each image $I_{t}$ in the sequence, the mean of the pixel intensities in the kidney region is computed using the PB-shape model as follows:(14)$${𝓂}_{t}=\frac{\sum_{\begin{array}{c} \left(x,y\right)\in \Omega \; \end{array}}^{\;}\;P_{K}\left(x,y\right)I_{t}\left(x,y\right)}{\sum_{\begin{array}{c} \left(x,y\right)\in \Omega \; \end{array}}^{}\;P_{K}\left(x,y\right)\;\;\;\;}$$

Note that this step does not require any kidney segmentation beforehand, thus can be carried out before starting our segmentation method. Figure 4 shows these mean values across the sequence in Figure 1. A number ℕ1 of images with the highest mean values (indicated by red circles in Figure 4) is selected to constitute the subset S1. Images of length ℕ2 whose 𝓂t values come next (indicted by black diamonds) are selected to form the subset S2. Finally, the remaining ℕ3 images (∑i=13ℕi=ℕ) in the sequence constitute the subset S3. Accordingly, the weight factor ωt in (6) is computed for each image in these subsets as follows:(15)ωt=1∀ It∈ S1ℕ2−𝒾/ℕ2∀ It∈ S2     0∀ It∈ S3
where 𝒾 is the index of the image It in S2, as decreasingly ordered by its 𝓂t value. The dashed green line in Figure 4 shows the values of ωt across the subject’s sequence shown in Figure 1.

Note that this partitioning procedure collects the high contrast images of the post-contrast interval of the MRI sequence in $\;S_{1}$, thus allowing the PB-shape model alone $\left(\omega_{t}=1\right)$ to accurately segment the kidneys from the $S_{1}$ images. The SS-shape model is constructed from the segmented kidneys from the $S_{1}$ images and used together with the PB-shape model (while $\omega_{t}$ is gradually decreasing) to segment the images in $S_{2}$. The SS-shape model is incrementally updated on the fly while working on the $S_{2}$ images (Figure 5). As a new segmentation becomes available, it is added onto the set employed that is to update the SS-shape model. Once all of the $S_{2}$ images are segmented, the SS-shape model is not updated anymore.

Note that the partitioning procedure keeps the more challenging, lower contrast images from the pre- and late-contrast intervals of the sequence in $S_{3}$. However, the SS-shape model is solely ($\omega_{t}=0$) able to precisely segment those $S_{3}$ images as it more accurately captures the kidney’s shape of this specific patient.

Finally, a flowchart of the proposed kidney segmentation method is shown in Figure 6.

## 3. Results

The performance of the proposed method was evaluated on DCE-MRI datasets of 45 subjects. The segmentation accuracy was assessed using DSC (mean ± standard deviation), IoU (mean ± standard deviation), and HD95 (mean ± standard deviation) metrics [2]. The PB-shape model was trained from the 30 ground-truth images of 30 different subjects. The parameters of the proposed method were experimentally set as follows: ε=1.5, λ1=6, λ2=6, ℕ1=20, ℕ2=10, and β=1. The values of all of the parameters were not changed or further tuned in all of the conducted experiments.

### 3.1. Method Performance with Comparisons to Other Methods

We first evaluated the performance of the proposed method on the gathered DCE-MRIs. Figure 7 depicts the segmentation process by our method for two different images. It shows the level set contour evolution during the segmentation procedure after different iterations. The figure also shows the final segmentation result. As shown in Figure 7, the proposed method can efficiently drive the contour towards the boundary of the kidneys in the images.

We then compared the segmentation performance of this new method against the following previous methods: FCMLS [19], FML [22], and PBPSFL [25]. In our experiments, we initialized—on purpose—the level set contour extremely far away from the kidney in all of the methods. We reported the performances on all of images and also on a particular set of low-contrast images (the first 5 images from each subject, totaling 225 images) in terms of DSC, IoU, and HD95 in Table 1.

The results in Table 1 demonstrate the improvement of the proposed method over our previous methods by achieving the highest mean DSC and IoU values and the lowest mean HD95 values, with a noticeable advantage on the low-contrast images. The lower standard deviation values of all of the evaluation metrics confirm the new method’s more consistent performance compared to the other methods. Figure 8 shows a qualitative comparison between these methods on two low-contrast images from two different subjects. Clearly, the proposed method achieves notably better segmentation accuracy than the other methods.

In order to further confirm the high-performance of the proposed method over the PBPSFL method [25], the two methods are used to segment the kidneys from the images that were corrupted by additive Gaussian noise (mean 0, variance 0.01, image intensities are normalized to range [0, 1]). Figure 9 visually compares the segmentation performances of both of the methods on a number of noisy images, while quantitative comparison results are given in Table 2.

The proposed method clearly outperforms the PBPSFL method in the presence of noise. It has a higher mean DSC and IoU, and lower mean HD95 values. While the two methods share the idea of using both the PB-shape and the SS-shape models, the higher performance of the new method can be attributed to the better shape model that was constructed, as explained in Section 2.5, and to updating the FCM memberships during the level set evolution.

The efficiency of the proposed method is further demonstrated by comparing its accuracy against those of a number of state-of-the-art level-set-based methods. Table 3 compares the accuracy of the results of all of the images by the proposed method and our previous FCMLS [19], PBPSFL [25], FML [22] methods, as well as by the shape-based (SB) method [33], the vector level set (VLS) [34], the 2nd-order MGRF level set (2nd-MGRF) [4], and a parametric kernel graph cut (PKGC) [35]. The DSC values of the PKGC and the 2nd-MGRF methods are reported in [5,6], using the same DCE-MRI datasets that were used in our study. As neither the output segmented kidneys that were obtained by these two methods nor the faithful implementations of the two methods are available to us, we are not able to compute/report the IoU and HD95 values of the two methods. Clearly, as shown in Table 3, the proposed method achieves the best segmentation accuracy compared with to other methods.

While we have not tried yet to optimize the time performance of the implementation of the new method, it takes about 8.4 min on the average to segment a sequence of 80 images with 256 × 256 sized pixels. However, the execution time of our previous method [25] segmenting the same sequence is 11.2 min. This demonstrates that the proposed method is faster than our previous method. All of the runtimes were calculated using MATLAB (R2015a) implementations of the methods on a 1.80 GHz Intel Core i7 CPU with 16 GB of RAM.

### 3.2. Ablation Experiments

We have performed an ablation study in order to assess the contribution of each component to the proposed method’s performance. We then evaluated the effect of some user-supplied parameters on the obtained segmentation accuracy. In our ablation study, we compared three scenarios. First, we evaluated the performance of our level-set-based method incorporating only fuzzy memberships and the PB-shape model. The FCM algorithm was used to compute the fuzzy memberships of the input image before the level set evolution began, and the memberships were not changed afterwards. In the second scenario, the FCM algorithm was embedded into the level set method, and the fuzzy memberships were updated as the level set evolved. The third scenario represented our complete approach, integrating the SS-shape model with the PB-shape model and the embedded fuzzy memberships. In all three of the scenarios, we assessed the impact on the final segmentation accuracy. The quantitative and qualitative comparison results are reported in Table 4 and Figure 10.

While the segmentation accuracy in Table 4 for all of the images improved from Scenario 1 to Scenario 2 to, eventually, Scenario 3, the impact was more prominent on low-contrast images. Updating the fuzzy memberships during the contour evolution improved the segmentation accuracy of the low-contrast images by about 3%, 4%, and 2.5 mm in terms of the mean DSC, IoU, and HD95, respectively. Moreover, the incorporation of the SS-shape model yielded a further improvement of about 3% in DSC, 5% in IoU, and 1.6 mm in HD95. As shown in Figure 10, the proposed method in Scenario 3 could more efficiently segment and catch the boundary of the target kidneys, thus generating more accurate segmentations. Overall, the results in Table 4 and Figure 10 highlight the benefit of the proposed integration of the embedded fuzzy memberships and the SS-shape model along with the PB-shape model into our level set framework.

Then, we studied the effect of some user-supplied parameters on the proposed method. We first investigated the impact of changing the values of ℕ1 and ℕ2 on the segmentation accuracy. Table 5 reports these results for three combinations of ℕ1 and ℕ2 values, demonstrating that the proposed method achieved the best segmentation accuracy when ℕ1=20 and ℕ2=10 as this allows more images to build the SS-shape model.

Afterwards, we evaluated the proposed method’s performance against different level set initializations. Figure 11 shows the accuracy of the proposed method on a sample DCE-MRI image with the level set contour that was initialized in different positions in the image. From the visual results and the reported DSC values in Figure 11, the segmentation accuracy was not changed in all cases. This confirms the method’s high and consistent performance, regardless of where the level set contour is initialized in the image.

### 3.3. Comparison to U-Net-Based Deep Neural Networks

In recent years, CNN in general, and the U-Net architecture in particular [13], have been applied to various medical image segmentation problems with good results [11,12,13,14,15,16,17,18]. Therefore, we have compared the proposed method versus a base U-Net CNN and one of its variants named BCDU-Net [28]. Both of the networks were trained from scratch on data from 18 subjects, were validated against data from 12 subjects, and were tested using the remaining 15 subjects. In order to prevent the models from overfitting, the data of each subject were augmented by performing the following operations on each image in the sequence: vertical and horizontal flipping, random x− and y-translations, rotation by ±45°, ±90°, 180° angles, and noising by adding Gaussian noise with zero mean and a variance of 0.01, 0.02, and 0.05 (the image intensities were normalized to range [0, 1]). The augmentation results in a total number of 16,404 images for training and 10,980 images for validation.

In order to further enlarge the data, following [18,36], we used high quality CT scans of 210 subjects from a KiTS19 dataset [29]. We manually split each image into two sub-images of size 256 × 256 for the left and right kidneys. This eventually increased the number of training and validation images to 40,050 and 10,980, respectively. The two networks were trained for 200 epochs using the Adam optimizer and an initial learning rate 0.0001, which decays by a factor of 0.1 whenever the validation loss is not reduced for 10 consecutive epochs. In order to further avoid overfitting, we used dropout regularization with a 50% ratio during the training. The training was carried out on a workstation with dual 2.20 GHz Intel Xeon Silver 4114 CPUs with 128 GB of RAM and two Nvidia GPUs in a Python environment, using Keras API with Tensorflow backend. The trained networks were then used to segment the kidney from the test subjects. Table 6 presents a comparison between the segmentation accuracies that were obtained by the proposed method, the U-Net model, and the BCDU-Net model with three dense blocks.

It can be seen from Table 6 that the BCDU-Net model had a considerably better performance than the base U-Net model, yet our method performed notably better than it. The mean DSC and IoU of the proposed method were higher than those of the U-Net and BCDU-Net models. Moreover, the mean HD95 values definitely showed the gap between the accuracy of our method and the two models. The HD95 metric characterized the divergence between the boundary surfaces of the segmentation result and the ground-truth kidney [2,27]. As such, unlike DSC, it was more sensitive to the shape deviations of the segmentation result against the ground-truth [27]. On the other hand, the standard deviations of DSC, IOU, and HD95 indicated that the proposed method was much more consistent and stable than the U-Net and BCDU-Net models. The improvement was more profound on the low-contrast images. It is important to mention that the behavior of the proposed method is easier to explain, as well as the interpretation of the results, compared to the deep U-Nets. For example, obtaining rather a noisy kidney contour from the segmentation result would suggest increasing the weighting factor $\lambda_{1}$ in our method, as a corrective action.

## 4. Conclusions

Kidney segmentation from DCE-MRI images is important for the assessment of renal transplant function. This paper has proposed a new and accurate method to automatically segment kidneys from DCE-MRI image sequences. The paper makes the following contributions:It integrates the FCM clustering algorithm, the level set method, and both PB-shape and SS-shape statistics for this problem *for the first time in literature*;The FCM clustering algorithm is embedded into the level set method; a pixel’s kidney/background fuzzy memberships are coupled with the level set evolution, considering the image intensities *directly,* as well as the kidney’s shape *indirectly*. This allows the proposed method to precisely capture the kidney, even on noisy and low-contrast images;The PB-shape and the SS-shape models are built using Bayesian parameter estimation, which statistically accounts for kidney pixels that are possibly not observed in the images that are used for the model building, thus rendering more accurate shape models;An automated, simple, and time-efficient strategy is proposed for partitioning the patient’s sequence into three subsets in order to properly determine the blending factor between the PB-shape and the SS-shape models;The experiments that were performed on 45 subjects demonstrate the accuracy of the proposed method and its robustness against noise, low contrast, and contour initialization *with no need for tuning the method’s parameters*. The comparisons with several state-of-the-art level set methods, and two CNN based on the U-Net architecture, confirm the *superior* and *consistent* performance of the proposed method.

Nevertheless, the proposed method has some limitations. First, incorporating the shape information into the level set method requires a prerequisite registration step in order to align an input DCE-MRI image to the shape prior model in order to compensate for the motion due to patient’s breathing and movement during the data acquisition. Any errors occurring in this alignment step would affect the segmentation performance. Second, similar to all of the level-set-based methods, the level set contour evolution in our method is guided by a partial differential equation containing several weighting parameters. Moreover, we use a weight factor that controls the contribution of the two shape statistics in the segmentation procedure. All of these weighting parameters require proper setting. Third, our new method takes about 7 s to segment one image of a size of 256 × 256 pixels, which is not suitable for real-time operation yet.

Our current research is directed towards improving the proposed method and alleviating its limitations. In our experiments, the values of the weighting parameters are experimentally chosen and then fixed throughout all of the conducted experiments without further tuning. We plan to investigate other weighting strategies in order to systematically find out the proper values for these weights, similarly to the scheme that was proposed in [37]. We also plan to investigate combining kidney segmentation and registration into the level set’s energy function. Simultaneously solving this issue for both of the tasks would diminish the propagation of errors from one task to the other. Last but not least, in an attempt to improve the time performance of the proposed method, we are working on converting the MATLAB code to C++ code that is optimized for GPU computing.

## Figures and Tables

**Figure 1 bioengineering-09-00654-f001:**
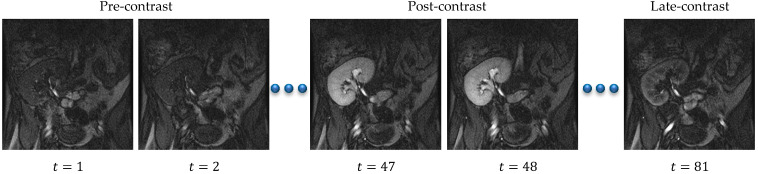
Contrast variation in DCE-MRI images of a patient’s kidney scanned at different time instants *t* after bolus injection. The concentration of the contrast agent in the kidney tissue is low at the beginning of the acquisition process, yielding low-intensity images (pre-contrast interval), reaches its maximum, generating high-intensity images (post-contrast interval), and then decreases slowly, resulting again in low-intensity images (late-contrast interval).

**Figure 2 bioengineering-09-00654-f002:**
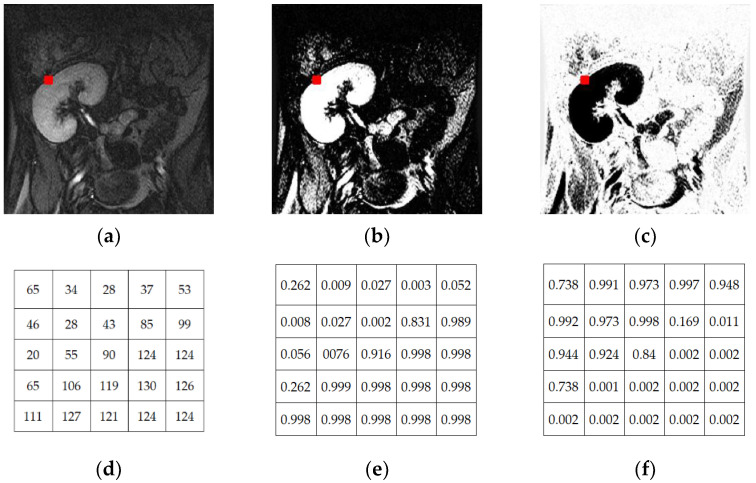
FCM clustering segmentation for a DCE-MRI grayscale image (**a**) into kidney cluster (**b**) and background cluster (**c**). The values of pixels in 5 × 5 windows centered at the red point are shown for the original DCE-MRI image in (**d**), the kidney cluster in (**e**), and the background cluster in (**f**), where $C_{K}=248.3$ and $C_{B}=96.2$.

**Figure 3 bioengineering-09-00654-f003:**
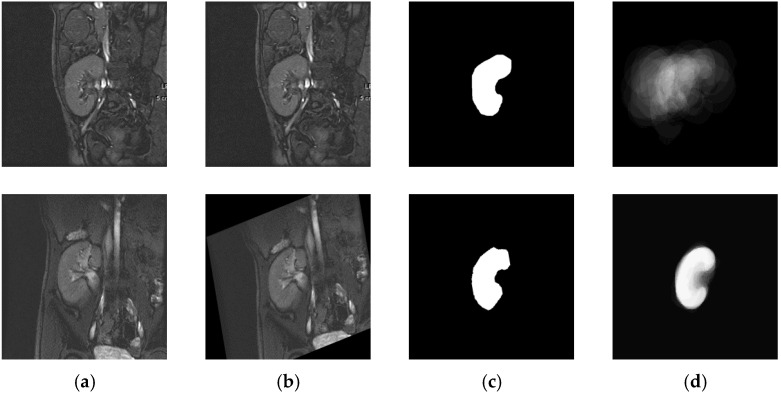
PB-shape model constructed using the Bayesian parameter estimation method: Some DCE-MRI kidney images before (**a**) and after (**b**) affine registration. Column (**c**) shows manually segmented kidneys after alignment. Column (**d**) shows the PB-shape model constructed before (top) and after (bottom) affine registration.

**Figure 4 bioengineering-09-00654-f004:**
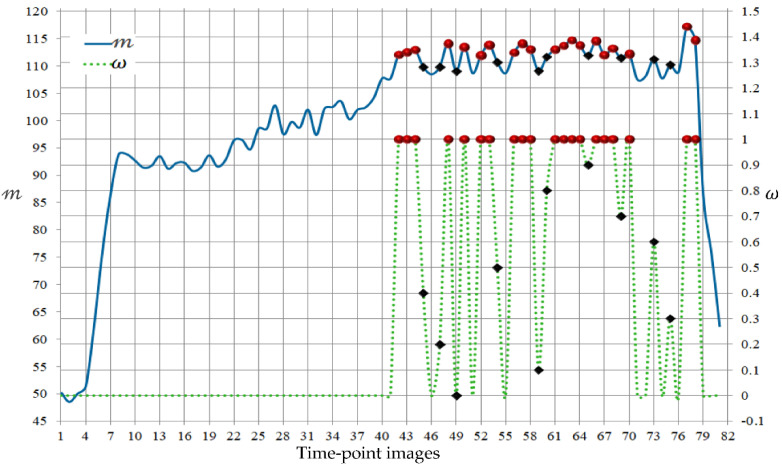
Changes of the mean of pixel intensities in the kidney region 𝓂t and weight factor ωt across the subject’s DCE-MRI sequence of Figure 1. Red circles indicate the highest contrast images included in subset S1 (ℕ1=20 ), while black diamonds refer to the next highest contrast images that comprise subset S2 (ℕ2=10 ).

**Figure 5 bioengineering-09-00654-f005:**
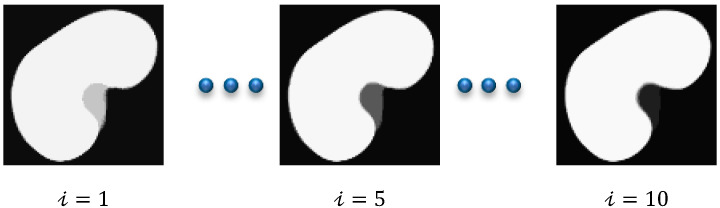
An SS-shape model constructed using Bayesian parameter estimation and updated during segmentation with S2 images of the subject’s sequence in Figure 1. As 𝒾 increases, the model more precisely captures the patient’s kidney shape (ℕ2 = 10).

**Figure 6 bioengineering-09-00654-f006:**
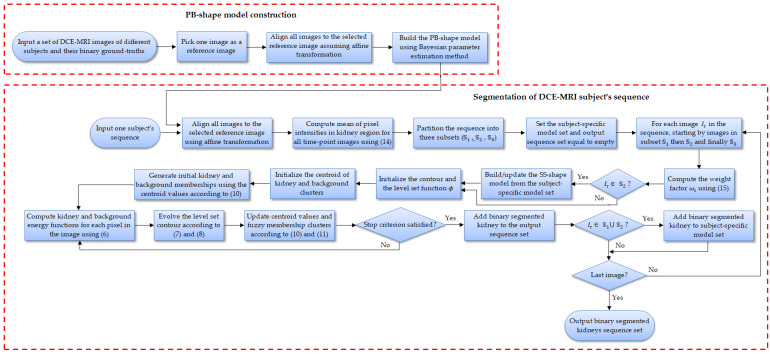
Flowchart of the proposed level-set-based kidney segmentation method.

**Figure 7 bioengineering-09-00654-f007:**
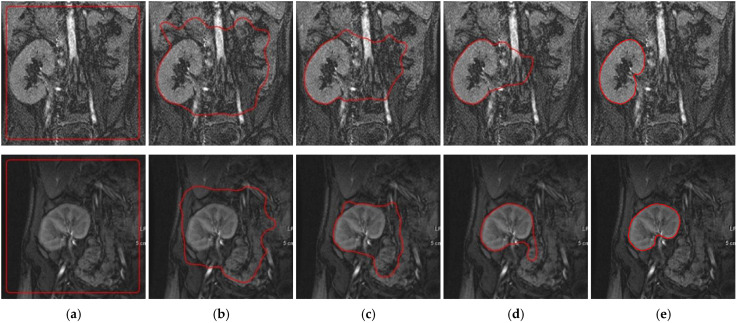
Evolution of the level set contour during the segmentation of the kidney from two DCE-MRI images (one per row) by the proposed method. (**a**) Initial level set contour. (**b**–**d**) Contour after 10, 30, and 40 iterations. (**e**) Final segmented kidneys obtained after 60 iterations.

**Figure 8 bioengineering-09-00654-f008:**
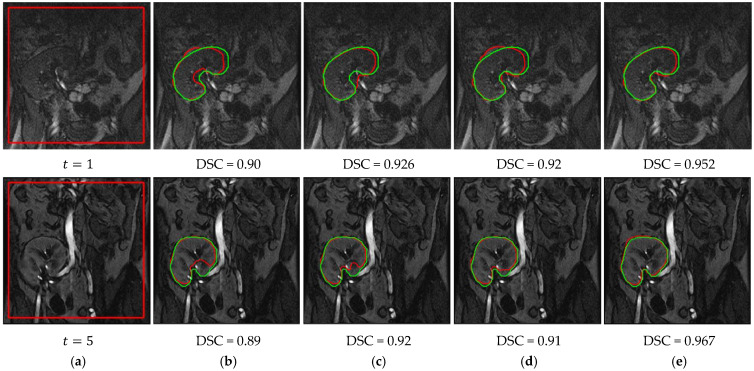
Segmentation results of the proposed method and our previous methods. (**a**) DCE-MRI kidney images with initial level set contour. Segmentation results (red outlines) with overlaid ground-truth segmentations (green outlines) along with corresponding DSC are shown for: (**b**) FCMLS method [19], (**c**) PBPSFL method [25], (**d**) FML method [22], and (**e**) proposed method.

**Figure 9 bioengineering-09-00654-f009:**
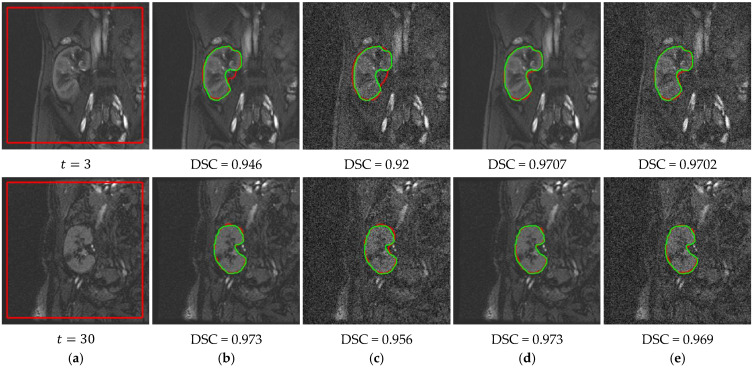
Performance of our method in the presence of noise compared with PBPSFL [25]. (**a**) DCE-MRI kidney images with added Gaussian noise and the initial level set contour are shown in red. Segmented kidneys are shown in red with DSC values extracted from original and noisy images by PBPSFL method [25] (**b**,**c**) and by the proposed method (**d**,**e**). Ground-truth segmentations are shown in green.

**Figure 10 bioengineering-09-00654-f010:**
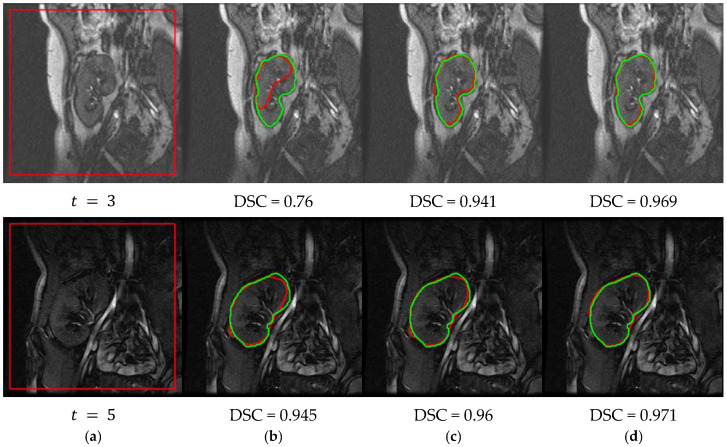
Segmentation results of two DCE-MRI images reflecting the effective role of each component in the proposed method. (**a**) DCE-MRI kidney images with the initial contour in red. Segmented kidneys in red, alongside their DSC values, obtained by our level-set-based method incorporating PB-shape model with (**b**) Fuzzy memberships, (**c**) Embedded fuzzy memberships, and (**d**) Embedded fuzzy memberships and the SS-shape model. Ground-truth segmentations are in green.

**Figure 11 bioengineering-09-00654-f011:**
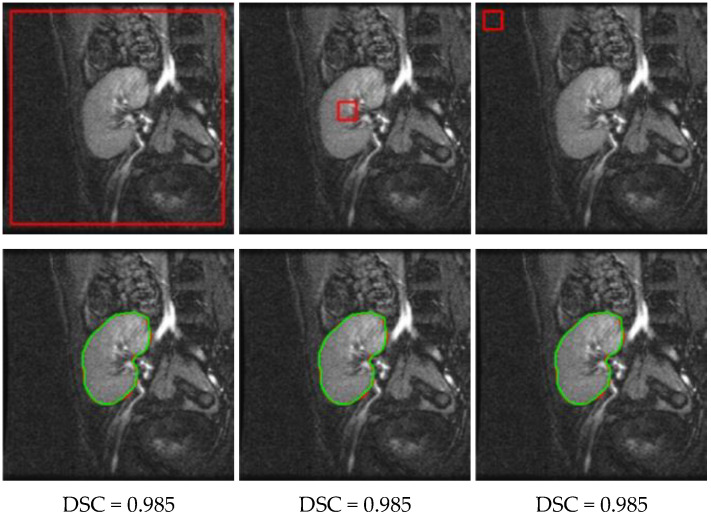
Segmentation results using the proposed method on a DCE-MRI image with different level set contour initializations (top row, red outlines). The segmented kidney (bottom row, red outlines), for any initialization, closely matches the ground-truth (green outlines), as evidenced by the associated DSC.

**Table 1 bioengineering-09-00654-t001:** Comparison between the segmentation performance of the proposed method and our previous methods.

Method	All Images			Low-Contrast Images		
	DSC	IoU	HD95	DSC	IoU	HD95
FCMLS [19]	0.941 ± 0.042	0.89 ± 0.056	1.78 ± 6.21	0.88 ± 0.137	0.80 ± 0.156	8.18 ± 22.8
PBPSFL [25]	0.952 ± 0.041	0.90 ± 0.043	1.11 ± 1.7	0.923 ± 0.13	0.88 ± 0.056	1.93 ± 2.32
FML [22]	0.956 ± 0.019	0.91 ± 0.035	1.15 ± 1.46	0.936 ± 0.024	0.88 ± 0.042	1.94 ± 1.58
Proposed	**0.953 ± 0.018**	**0.91 ± 0.033**	**1.10 ± 1.4**	**0.942 ± 0.02**	**0.90 ± 0.034**	**1.56 ± 1.46**

**Table 2 bioengineering-09-00654-t002:** Comparison between the segmentation performance of the proposed method and the PBPSFL Method on noisy images.

Method	All Images			Low-Contrast Images		
	DSC	IoU	HD95	DSC	IoU	HD95
PBPSFL [25]	0.944 ± 0.022	0.89 ± 0.039	1.71 ± 1.7	0.93 ± 0.025	0.87 ± 0.042	2.47 ± 1.85
Proposed	**0.952 ± 0.016**	**0.91 ± 0.029**	**1.20 ± 1.0**	**0.95 ± 0.018**	**0.90 ± 0.033**	**1.41 ± 1.24**

**Table 3 bioengineering-09-00654-t003:** Comparison between the segmentation accuracy of the proposed method and the existing methods.

Method	DSC	IoU	HD95
PKGC [35]	0.820 ± 0.180	-	-
VLS [34]	0.902 ± 0.083	0.84 ± 0.12	3.62 ± 7.29
SB [33]	0.912 ± 0.043	0.84 ± 0.07	2.64 ± 1.63
FCMLS [19]	0.941 ± 0.042	0.89 ± 0.056	1.78 ± 6.21
2nd-MGRF [4]	0.943 ± 0.028	-	-
PBPSFL [25]	0.952 ± 0.041	0.90 ± 0.043	1.10 ± 1.69
FML [22]	0.956 ± 0.019	0.91 ± 0.035	1.15 ± 1.46
Proposed	**0.953 ± 0.018**	**0.91 ± 0.033**	**1.1 ± 1.4**

**Table 4 bioengineering-09-00654-t004:** Ablation study: Segmentation performance of the proposed method in the three scenarios.

Method	All Images			Low-Contrast Images		
	DSC	IoU	HD95	DSC	IoU	HD95
PB-shape + Fuzzy memberships	0.945 ± 0.055	0.89 ± 0.056	1.63 ± 3.87	0.884 ± 0.12	0.81 ± 0.128	5.61 ± 12.54
PB-shape + Embedded fuzzy memberships	0.946 ± 0.029	0.89 ± 0.048	1.63 ± 1.97	0.918 ± 0.06	0.85 ± 0.096	3.18 ± 4.28
PB-shape + Embedded memberships + SS-shape	**0.953 ± 0.018**	**0.91 ± 0.033**	**1.10 ± 1.4**	**0.942 ± 0.02**	**0.90 ± 0.034**	**1.56 ± 1.46**

**Table 5 bioengineering-09-00654-t005:** Segmentation performance of the proposed method for different ℕ1 and ℕ2 values.

Experiment	ℕ1	ℕ2	All Images			Low-Contrast Images		
			DSC	IoU	HD95	DSC	IoU	HD95
1	15	15	0.949 ± 0.021	0.90 ± 0.038	1.34 ± 1.43	0.942 ± 0.022	0.89 ± 0.038	1.58 ± 1.46
2	20	10	**0.953 ± 0.018**	**0.91 ± 0.033**	**1.10 ±** **1.4**	**0.942 ± 0.02**	**0.90 ± 0.034**	**1.56 ± 1.46**
3	10	20	0.946 ± 0.027	0.89 ± 0.038	1.41 ± 1.62	0.94 ± 0.023	0.88 ± 0.041	1.61 ± 1.48

**Table 6 bioengineering-09-00654-t006:** Comparison between the segmentation performance of the proposed method versus U-Net and BCDU-Net models.

Method	All Images			Low-Contrast Images		
	DSC	IoU	HD95	DSC	IoU	HD95
U-Net [13]	0.940 ± 0.041	0.89 ± 0.069	10.30 ± 23.8	0.88 ± 0.071	0.77 ± 0.13	19.9 ± 28.8
BCDU-Net [28]	0.942 ± 0.038	0.89 ± 0.062	4.62 ± 12.35	0.90 ± 0.057	0.82 ± 0.089	7.89 ± 12.27
Proposed	**0.957 ± 0.016**	**0.93 ± 0.019**	**0.80 ± 1.03**	**0.952 ± 0.014**	**0.90 ± 0.026**	**0.85 ± 0.76**

## Data Availability

Data are available upon reasonable request to the corresponding author.

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
