# Peer review of "Level-Set-Based Kidney Segmentation from DCE-MRI Using Fuzzy Clustering with Population-Based and Subject-Specific Shape Statistics"

_bioengineering, 2022, doi:10.3390/bioengineering9110654_

Round 1

Reviewer 1 Report

The study is an interesting one. I have only a few suggestions:

1. Please correct section 0 in the structure of the paper at the last paragraph of introduction section.

2.  There are some 'Error!Reference source not found. ' errors in the manuscript. Please correct them.

3. Why U-Net  is just used as the backbone network to extract image feature maps? Why other well-known backbone networks (e.g., DenseNet, VGGNet, EfficientNet, etc.,) are not used?

4. More technical research gaps and future research directions should be provided.

5. Please highlight the most important advantages and disadvantages of the proposed methodology.

6. Can other metrics be applied to evaluate the segmentation model (like Intersection-Over-Union (Jaccard Index))?

7. Can the different kidney datasets be mentioned in literature review with their features?

Author Response

The authors like to thank the reviewer for the critical and constructive comments, which for sure have improved the content and presentation of the manuscript. Below we provide point-to-point responses to the reviewer's comments and concerns. The relevant parts in the revised manuscript are highlighted in yellow.  

1.1 Concern: Please correct section 0 in the structure of the paper at the last paragraph of introduction section.

Response:

Thanks a lot for pointing this out. We have fixed this issue in the revised manuscript.

1.2 Concern: There are some 'Error!Reference source not found. ’ errors in the manuscript. Please correct them.

Response:

Thank you for pointing this out. We have fixed all reference errors in the revised manuscript.

1.3 Concern: Why U-Net is just used as the backbone network to extract image feature maps? Why other well-known backbone networks (e.g., DenseNet, VGGNet, EfficientNet, etc.,) are not used?

Response:

Thanks for this question. The method we are proposing is based on fuzzy clustering and level set methods. We compare its performance against several existing methods in the literature including non-deep-learning-based methods (e.g., existing level set-based methods) as well as deep learning methods.  Extensive work has been done for kidney segmentation from computed tomography images using deep neural networks. However, the number of studies attempted to perform kidney segmentation in MRI (which is our case in this manuscript) is limited. We review these methods [11-18] in the revised manuscript on Pages 2-3.  Several additional related references have been added to our reference list in the revised manuscript. Our choice of the U‑Net model [13] and its variations is motivated by their choice of the various researchers in these related works [11-18].  Moreover, U‑Net model and its variations have shown great success in the medical field, especially for image segmentation.

As such, we use the original 2D U-Net model [13] in our manuscript for the sake of comparison with our proposed level set method. Moreover, motivated by the reviewer’s question, we investigate in the revised manuscript an additional deep model for the same task. We investigate using the BCDU-Net model [28] as one of the successful variations of the U-Net model. The reader is referred to [28] for more details on this model architecture.

Both deep networks, the original U-Net model [13]  and the BCDU-Net model [28],  are trained on a workstation with dual 2.20GHz Intel Xeon Silver 4114 CPUs with 128GB of RAM and two Nvidia GPUs in a Python environment using Keras API with Tensorflow backend. The models are trained for 200 epochs using Adam optimizer with an initial learning rate 0.0001. The learning rate is decayed by a factor of 0.1 whenever the validation loss is not decreased for 10 consecutive epochs. To further avoid overfitting, dropout regularization is applied during network training. We have added this investigation in Section 3.3 on Pages 15-16 in the revised manuscript.

Table 6 on Page 16 in the revised manuscript gives a comparison between the results obtained by the proposed method versus the U-Net and BCDU-Net results on all images and on the low‑contrast images. It is clear from this comparison that the BCDU-Net model performs notably better than the original U-Net model. More importantly, the proposed method performs significantly better than both the U-Net and BCDU-Net models on segmenting kidneys from high and low‑contrast images.

1.4 Concern: More technical research gaps and future research directions should be provided.

Response:

Thanks for this suggestion. We have re-structured the introduction section in the revised manuscript. We have added a paragraph on Pages 2-3 in the introduction to survey the state-of-the-art methods for kidney segmentation from MRI using deep neural networks. Several additional related references have been added to our reference list in the revised manuscript.

Moreover, we have dedicated a part of the introduction section to “Research Gap” (with side heading) on Page 3, in which we explain the shortcomings and limitations of the existing methods. In this part, we explain that the level set (LS)-based methods developed in the literature may achieve high performance in segmenting the kidney from its surroundings. However, the majority of these methods require an accurate delineation for the initial contour which is performed manually by the user. Lack of good initialization may result in the method’s complete failure or a deficit in the resulting accuracy.  Moreover, several of these methods suffer on low-contrast images like the ones in the pre-contrast and late-contrast intervals of the sequence in Fig. 1. On the other hand, lacking a sufficiently large number of annotated training data, the deep network-based methods in the literature could not achieve high segmentation accuracy. Then we show in the remaining part of the introduction how our proposed method helps alleviate these shortcomings.

We report several ongoing and future research directions at the end of the conclusion section on Page 17 of the revised manuscript. These research directions are aimed to improve the method’s accuracy, ease-of-use and time performance.

1.5 Concern: Please highlight the most important advantages and disadvantages of the proposed methodology.

Response:

Thanks for this suggestion. We first give the advantages of the proposed methodology in the introduction section on Pages 3-4 of the revised manuscript. Then to stress more the advantages, we enumerate them in the Conclusions section on Pages 16-17 of the revised manuscript.

Then we discuss some limitations of the proposed method in the Conclusions section on Page 17. This paves the way to end this section with some ongoing and future research efforts to improve the proposed method and overcome these limitations.

1.6 Concern: Can other metrics be applied to evaluate the segmentation model (like Intersection-Over-Union (Jaccard Index))?

Response:

Thanks a lot for this suggestion. The accuracy of the proposed method is assessed through the comparison of the method’s outputs with the ground-truth segmentations provided by the expert radiologists. The accuracy is objectively quantified in terms of two popular evaluation metrics, namely, Dice similarity coefficient and 95% percentile of Hausdorff distance [2, 27]. Per the reviewer’s suggestion, we add a third metric, the Intersection-Over-Union (IoU) metric. We have updated the manuscript to report the three metrics in all our results (see Tables 1-6 in the Results Section of the revised manuscript). All three metrics confirm the superior performance of the proposed method.

 1.7 Concern: Can the different kidney datasets be mentioned in literature review with their features?

Response:

Thanks a lot for this suggestion. We are aware of two different kidney datasets that are recently released in the public domain. The first dataset is published by the CHAOS challenge [18]. This dataset includes data of 80 different subjects, 40 CTs and 40 MRIs. Each sequence contains an average of 90 scans in CT and 36 in MRI in DICOM format. The second dataset is KiTS19 challenge dataset [29]. It contains 300 subjects’ data, 210 out of all data are publicly released for training and the remaining 90 subjects are held out for testing. Each subject has a sequence of high quality CT scans with their ground-truth labels that are manually segmented by medical students. It also includes a chart review that illustrates all relevant clinical information about this patient. All CT images and segmented annotations are provided in an anonymized NIFTI format.

Information about the two datasets are added at the end of the fifth paragraph of Introduction Section on Page 3 and the second-to-last paragraph of Introduction Section on Page 4 in the revised manuscript.

Reviewer 2 Report

General Comments

Reviewed is the manuscript “Level Set-Based Kidney Segmentation from DCE-MRI Using Fuzzy Clustering with Population-Based and Subject-Specific Shape Statistics” submitted by Moumen El-Melegy, et, al. The authors proposed a DCE-MRI kidney segmentation method that combining fuzzy c-means clustering with iteratively update of fuzzy membership. The article is well organized with a smooth flow of information during the explanation of each method. It is well-written, with very few clerical errors, and the style and layout are very well articulated. Overall, the authors clearly demonstrate their approach and detail the performance gained in this research field and the article meets the required standards for publication after minor edits.

Specific Comments

-           Add more details to figure 1-5 description.

-           In figure 1, how to show the clusters?

-           In figure 2, top figures of panel A and panel B looks very similar.

-           In the paragraph after table 6, please fix the reference issues “From Error! Reference source not found.,”

-           It's important to share the code of the experiments to guarantee the reproducibility of the experiments and to assure the veracity of the results.

Author Response

The authors like to thank the reviewer for the critical and constructive comments, which for sure have improved the content and presentation of the manuscript. Below we provide point-to-point responses to the reviewer's comments and concerns. The relevant parts in the revised manuscript are highlighted in yellow.

2.1 Concern: Add more details to figure 1-5 description.

Response:

Thank you for pointing this out. We have updated all the figures' captions in the revised manuscript to better explain their contents.

2.2 Concern: In figure 1, how to show the clusters?

Response:

Thanks for your question. As mentioned in the first paragraph of Introduction Section in our manuscript, DCE-MRI data acquisition is done through injecting the patient with a contrast agent and, during the perfusion; kidney images are captured quickly and repeatedly at 3 sec intervals using a 1.5T MRI scanner. The contrast agent perfusion leads to contrast variation in the acquired images. Consequently, the intensity of images at the beginning of the sequence is low (pre-contrast interval), gradually increases until reaching its maximum (post‑contrast interval), and then decreases slowly (late-contrast interval). In Figure 1, we show a time sequence of DCE‑MRI kidney images of one of the patients taken during the contrast agent perfusion. We have added more details in the figure caption to better explain its contents.

2.3 Concern: In figure 2, top figures of panel A and panel B looks very similar.

Response:

Thank a lot for pointing this out. Figure 2 aims to illustrate the segmentation results of the FCM clustering algorithm for a DCE-MRI kidney image. Figure 2(a) shows the input of a grayscale kidney image.  Figure 2(b) shows the obtained kidney cluster while Figure 2(c) shows the obtained background cluster. Each cluster is illustrated by displaying the fuzzy memberships (that are in the range ) of all image pixel in this specific cluster. The membership values of pixels to a specific cluster depend on the distances between the intensity of pixels and the centroid value of the cluster. This means that pixels are assigned high membership values (close to 1) to a certain cluster when their intensities are close to the centroid value and low membership values (close to 0) when they are far from the centroid. That is, the more the brightness of a pixel in Panel B of the figure, the higher its probability belonging to the kidney cluster. Similarly, the more the brightness of a pixel in Panel C of the figure, the higher its probability belonging to the background cluster. We briefly explain this in the last paragraph of Section 2.4 on Page 6 in the revised manuscript. In addition, in order to further clarify this, we update Figure 2 on Page 7 in the revised manuscript to show zoom-in windows of the original image and the obtained memberships at the indicated red point. 

2.4 Concern: In the paragraph after table 6, please fix the reference issues “From Error! Reference source not found.,”

Response:

Thank you for pointing this out. We have fixed all reference errors in the revised manuscript.

2.5 Concern: It's important to share the code of the experiments to guarantee the reproducibility of the experiments and to assure the veracity of the results.

Response:

Thanks a lot for this suggestion. The code of the current study is available from the authors upon reasonable request and with permission of the sponsor of this work, STDF. 

Reviewer 3 Report

(1)Please check the manuscript carefully to remove the typos, improve the language and format. E.g.

- Error! Reference source not found.

(2)Please provide and label the reference indices of the compared methods in the figures and tables, such as Table 3, and then the readers can judge whether the compared methods are SOTA.

(3)Please use bold font to label the best results in all tables, such as Table 1.

(4)Why are the compared methods different in the tables and figures? All the comparisons should be fair, objective and comprehensive rather than biased, subjective and selective.

(5)Since weight factor is used in this paper, the authors could introduce and cite some weighting strategies, such as:

-Dynamic weighted discrimination power analysis: a novel approach for face and palmprint recognition in DCT domain. International Journal of the Physical Sciences

-Dual-source discrimination power analysis for multi-instance contactless palmprint recognition

(6)The authors should plot a procedure framework of the proposed method. It is better to show the result of each step.

Author Response

The authors like to thank the reviewer for the critical and constructive comments, which for sure have improved the content and presentation of the manuscript. Below we provide point-to-point responses to the reviewer's comment and concerns. The relevant parts in the revised manuscript are highlighted in yellow.  

3.1 Concern: Please check the manuscript carefully to remove the typos, improve the language and format. E.g.- Error! Reference source not found.

Response:

Thanks a lot for pointing this out. We have fixed all typos and reference errors in the revised manuscript. We have also done our best to correct any English mistakes and improve the format of the revised manuscript.

3.2 Concern: Please provide and label the reference indices of the compared methods in the figures and tables, such as Table 3, and then the readers can judge whether the compared methods are SOTA.

Response:

Thanks a lot for this suggestion. We have added references for all comparison methods in all tables and figures in the revised manuscript.

3.3 Concern: Please use bold font to label the best results in all tables, such as Table 1.

Response:

Thanks a lot for this suggestion. We have marked the best results in Tables 1-6 in the revised manuscript in bold.

3.4 Concern: Why are the compared methods different in the tables and figures? All the comparisons should be fair, objective and comprehensive rather than biased, subjective and selective.

Response: 

Thanks a lot for this question. In our experiments, we have followed this evaluation/assessment strategy:

We first compare the segmentation performance of our method with those of our previous methods, namely FCMLS [19], FML [22], and PBPSFL [25]. We provide quantitative and qualitative comparison results in Table 1 and Figure 7 in our manuscript. Then, we further demonstrate the advantages of the propose method over our earlier PBPSFL method [25] on which we build and develop our current method. We compare their performances in the presence of noise and we present quantitative and qualitative comparison results in Table 2 and Figure 8 in our manuscript.

Having demonstrated that the proposed method outperforms all our own previous methods, we compare its performance against several state-of-the-art level set-based methods (including our own pervious methods) in Table 3 in the manuscript.

In Tables 4 and 5, we report the results of our ablation experiments to investigate the impact of the different components in the proposed method and the effect of some method’s parameters on its obtained accuracy. Figure 10 shows the segmentation results using the proposed method with different level set contour initializations confirming the consistent performance of the method regardless of where the initialization has started.

Eventually, we dedicate Section 3.3 in our experimental results to compare the proposed methods against two deep neural networks. Table 6 on Page 16 in the revised manuscript gives a comparison between the results obtained by the proposed method versus the U-Net and BCDU-Net results on all images and on the low‑contrast images. Clearly, the proposed method performs significantly better than both the U-Net and BCDU-Net models on segmenting kidneys from high and low‑contrast images.

3.5 Concern: Since weight factor is used in this paper, the authors could introduce and cite some weighting strategies, such as:

-Dynamic weighted discrimination power analysis: a novel approach for face and palmprint recognition in DCT domain. International Journal of the Physical Sciences

-Dual-source discrimination power analysis for multi-instance contactless palmprint recognition

Response:

Thanks a lot for referring us to these references. We do agree. As almost all the existing level set-based methods, our new method includes some weighting parameters to reflect the contributions of the different components in the partial differential equation governing the level set evolution. Moreover, we use a weight factor that controls the contribution of both shape statistics in the segmentation procedure. The values of these parameters are experimentally chosen after performing several preliminary experiments, and then are not changed or further tuned throughout all conducted series of experiments. However, as the reviewer suggested, we can use other weighting strategies to systematically find out proper values for these weights, like the one proposed in the paper by Leng et al in the International Journal of Physical Sciences.  

We note this issue in the Conclusions Section on Page 17 in the revised paper in the context of discussing the proposed method’s limitations and how to overcome them in our ongoing and future research efforts. The above reference by Leng et al has been added to our reference list.

3.6 Concern: The authors should plot a procedure framework of the proposed method. It is better to show the result of each step.

Response:

Thanks a lot for this suggestion. We have added two algorithms at the end of Section 2.5 on Page 8 and Section 2.6 on Page 10 in the revised manuscript. Algorithm 1 explains the steps of constructing the PB-shape information model, which is done offline from a set of ground-truth kidney segmentations (done manually by expert radiologists) from different subjects. Algorithm 2 explains the steps of the proposed kidney segmentation method, which is run on the subject’s sequence to be segmented.

Furthermore, per the reviewer’s suggestion, we add Figure 6 in Section 3.1 on Page 11 in the revised manuscript. This figure illustrates the segmentation process of our proposed method on two different images. It shows the level set contour evolution during the segmentation procedure after different iterations. Eventually, the figure shows the final segmentation result. As apparent in Figure 6, the proposed method can efficiently drive the contour towards the boundary of kidneys in the images.

Round 2

Reviewer 3 Report

(1)Please plot a flowchart of the whole method in Section 2.

(2)Why are some results missing in Table 3? Please clearly state the reason in the text.

Author Response

The authors like to thank the reviewer for his critical and constructive comments, which for sure have improved the content and presentation of the manuscript. Below we provide point-to-point responses to the reviewer’s comments and concerns. The relevant parts in the revised manuscript are highlighted in yellow.  

1.1 Concern: Please plot a flowchart of the whole method in Section 2.

Response:

Thanks for this suggestion. We add Figure 6 at the end of Section 2 on Page 10 in the revised manuscript. This figure shows the flowchart of the proposed method.

1.2 Concern: Why are some results missing in Table 3? Please clearly state the reason in the text

Response:

Thanks for this question. The authors in [5, 6] investigated the performance of PKGC [35] and 2nd-MGRF [4] methods in the context of kidney segmentation from the same DCE-MRI datasets used in our study. They only used the DSC metric to measure the segmentation accuracy of these methods, while we assessed the performance of all methods in our manuscript using DSC, IoU, and HD95 metrics. Thus, we report the DSC values in Table 3 as reported in [5, 6]. As neither the output segmented kidneys obtained by these two methods nor faithful implementations of the two methods are available to us, we are not able to compute/report IoU and HD95 values of the two methods. However, the DSC values reported in Table 3 clearly show the better performance of the proposed method over the PKGC and 2nd-MGRF methods. This information is added to the revised manuscript on Page 12, commenting on Table 3.